# Exploring maintenance of physical activity behaviour change among people living with and beyond gastrointestinal cancer: a cross-sectional qualitative study and typology

Chloe Grimmett ,[1] Claire Foster,[1] Katherine Bradbury,[2] Phillippa Lally,[3] Carl R May,[4] Michelle Myall ,[5] Bernardine Pinto,[6] Teresa Corbett[5]

► Prepublication history and supplemental material for this paper is available online. To view these files, please visit the journal online (http://dx.doi.org/10.1136/bmjopen-2020-037136).

For numbered affiliations see end of article.

**Correspondence to**
Dr Chloe Grimmett;
C.Grimmett@soton.ac.uk

## ABSTRACT

**Objectives** In the last decade, there has been a rapid expansion of physical activity (PA) promotion programmes and interventions targeting people living with and beyond cancer (LWBC). The impact that these initiatives have on long-term maintenance of PA remains under-researched. This study sought to explore the experiences of participants in order to characterise those who have and have not successfully sustained increases in PA following participation in a PA intervention after a diagnosis of gastrointestinal (GI) cancer, and identify barriers and facilitators of this behaviour.

**Design** Cross-sectional qualitative study. Semi-structured interviews with participants who had previously taken part in a PA programme in the UK, explored current and past PA behaviour and factors that promoted or inhibited regular PA participation. Interviews were audio-recorded, transcribed verbatim and analysed using thematic analysis. Themes and subthemes were identified. Differences between individuals were recognised and a typology of PA engagement was developed.

**Participants** Twenty-seven individuals (n=15 male, mean age=66.3 years) with a diagnosis of GI cancer who had participated in one of four interventions designed to encourage PA participation.

**Setting** UK.

**Results** Seven themes were identified: disease processes, the role of ageing, emotion and psychological well-being, incorporating PA into everyday life, social interaction, support and self-monitoring and competing demands. A typology with three types describing long-term PA engagement was generated: (1) maintained PA, (2) intermittent PA, (3) low activity. Findings indicate that identifying an enjoyable activity that is appropriate to an individual's level of physical functioning and is highly valued is key to supporting long-term PA engagement.

**Conclusion** The typology described here can be used to guide stratified and personalised intervention development and support sustained PA engagement by people LWBC.

## INTRODUCTION

There is robust evidence that physical activity (PA) can provide improvements in

### Strengths and limitations of this study

► This qualitative study using in-depth interviews with individuals engaging in varying levels of physical activity provides novel insight into the barriers and facilitators to long-term physical activity (PA) engagement after diagnosis and treatment of cancer.

► The typology presented can help inform the design and delivery of personalised PA interventions to support long-term behaviour change.

► Findings may be limited because participants were recruited from completed PA interventions and therefore do not reflect the views and experiences of individuals who were unwilling or unable to take part in such a trial.

physiological and psychosocial outcomes in people living with and beyond cancer (LWBC), including enhanced quality of life, physical function and reductions in cancer-related fatigue.[1] Furthermore, observational data suggest PA may reduce cancer recurrence, death from cancer and other causes in some cancer types. Therefore, cancer survivors potentially stand to benefit considerably from engagement in PA, and consequently, it is recommended that regular PA be promoted as part of the cancer care pathway.[2] The most recent guidelines from the American College of Sports Medicine suggest those LWBC should avoid inactivity and aim to achieve public health guidelines of 150 minutes per week of aerobic exercise and twice a week resistance exercise. This level of PA can also reduce the risk of developing chronic diseases, delay decline in physical and mental functioning associated with ageing, and extend life.[3]

The majority of cancer survivors do not meet these recommendations and are less

active than those without a history of cancer.[4] There has been a rapid expansion of PA promotion programmes and interventions targeting people LWBC to address this imbalance. Evidence suggests on-site supervised, home-based, web-based, individual and group and peer-support interventions can successfully increase PA in the short and medium term.[5–7] Furthermore, a recent systematic review and meta-analysis of long-term PA behaviour reported a moderate impact at least 3 months post-intervention completion.[8] However, there is wide variation of impact at an individual level with large standard deviations in self-reported and objectively recorded PA. This is elegantly illustrated by Morey and colleague's trajectory analysis of the RENEW study, a broad-reach intervention delivered by printed materials and telephone to older adults with a diagnosis of cancer.[9] The authors report that a small proportion (7%) of participants did not engage in any PA at any point during the study. This is in contrast to 33% who increased PA at the end of the intervention but declined during the observational follow-up and around 60% who maintained increases in PA during the intervention and 12-month follow-up period. If interventions are to be developed that successfully support as many individuals as possible to maintain a physically active lifestyle, we need to understand factors that promote and inhibit this maintenance of PA.

Individuals diagnosed with gastrointestinal (GI) cancers, such as colon, rectal, stomach and oesophageal, are a particularly important group to consider. Affecting both men and women, cancers of the colon and rectum are particularly prevalent in older age, which means individuals frequently present with other comorbidities. Individuals LWB GI cancer may also experience troublesome symptoms such as stool frequency and urgency, incontinence, fatigue and pain,[10] making adoption and maintenance of PA particularly challenging.

There is a paucity of published literature exploring challenges and enablers to long-term PA behaviour change among cancer populations. A recent mixed methods review investigated barriers, facilitators and preferences for PA among people LWBC and found that common barriers were treatment-related side effects, lack of time and fatigue.[11] Facilitators included exercising to gain control over health and feelings of well-being. However, the majority of studies in this review focused on uptake and/or participation in interventions, with little attention paid to experiences of independent and sustained (or not) behaviour change postintervention. Furthermore, only 4 of the 19 studies included individuals with GI cancer.

This paper reports findings from a qualitative study that sought to explore the experiences of participants in order to characterise those who have and have not successfully sustained increases in PA following participation in a PA intervention after a diagnosis of GI cancer, and identify factors that promote and inhibit this behaviour.

## METHODS
### Recruitment and procedure
Participants were recruited from four PA promotion programmes in England and Northern Ireland (see table 1). The coordinators of the programmes identified eligible participants: previous diagnosis of colorectal, oesophageal or stomach cancer and had completed the PA programme more than 6 months before participating in the current study. A letter of invitation to take part in the current study was sent by post. Those who were willing were asked to return a reply slip to the coordinating centre at the University of Southampton.

| Table 1 | Programmes from which participants were recruited |
|---|---|
| **Project** | **Description** |
| The Advancing Survivor Cancer Outcomes Trial (ASCOT) | A lifestyle programme for those diagnosed with breast, prostate or colorectal cancer. ASCOT was a distance-based lifestyle intervention, delivered through printed materials and personally tailored telephone discussions underpinned by habit theory.[29] See published protocol.[30] Identified participants had a previous diagnosis of colorectal cancer and had set at least one PA goal during the study. |
| Efficacy of aN exercise training programme during Concurrent neoadjuvant canceR treAtments trial (ENCOURAGE) | ENCOURAGE participants with oesophageal and stomach cancer took part in a structured, supervised hospital-based exercise programme before during and after cancer treatments and prior to surgery. |
| Active Everyday | A PA referral programme based in Sheffield, UK. Participants received personalised behaviour change support and sign-posting to local PA opportunities.[31] Participants with a previous diagnosis of colorectal cancer were identified. |
| Macmillan Move More Programme – Northern Ireland | Participants receive personalised support with one-to-one consultations and advice to increase their activity levels and appropriate sign-posting to local opportunities including group-based exercise programmes for people LWBC. Participants with a previous diagnosis of cancer were identified. |

LWBC, living with and beyond; PA, physical activity.

### Ethical statement

Participants were provided with information sheets and gave written consent prior to interview.

### Procedure

The study was reported according to the Consolidation Criteria for Reporting Qualitative Research (COREQ).[12] See online supplemental file 1.

All interviews were conducted by CG (an experienced qualitative researcher) and audio-recorded (with consent) using an encrypted electronic device. CG is a member of the trial steering committee for the ASCOT study but did not have direct involvement with the delivery of the intervention. Participants were informed that the current study was independent of the study in which they had previously taken part.

One-to-one telephone interviews took place between January and June 2019 and lasted between 40 and 75 min. Semi-structured interviews were carried out using a topic guide comprising open questions designed to explore and capture data on factors that influenced the maintenance (or not) of PA following participation in a PA programme (see online supplemental file 2).

Participants were also asked to complete a short demographics questionnaire to capture, age, sex, cancer type, marital status, ethnicity, level of education, home and car ownership, occupation and caring responsibilities.

### Analysis

A thematic analysis was conducted as described by Braun and Clarke.[13] A recursive process was employed, moving back and forth between stages with continuous interpretation of the data from the outset.

(1) Transcription: interviews were audio-recorded using an encrypted audio recorder and transcribed verbatim by an independent, experienced transcriber. Transcripts were checked against audio files, were anonymised and pseudonyms attributed. (2) Familiarisation with data: transcripts and field notes were read and re-read with analytical notes, thoughts and impressions recorded freehand. (3) Generating initial codes: transcripts were read line by line and initial codes applied. TC, CG and CRM coded a sample of transcripts (n=3). A data analysis workshop was then held to discuss codes and their labels. TC and CG coded a further three transcripts followed by a second analysis workshop before CG coded the remaining interviews. (4) Searching for themes: codes were then sorted into themes and subthemes by CG and an initial thematic map generated. (5) Reviewing, defining and naming themes: themes were data driven, identified at a semantic level within the realist paradigm. A further analysis workshop was held between TC, CG and CRM to discuss the themes, subthemes and the relationships between them, while also interrogating the data between cases. Refinements were made and CG revisited the transcripts to determine if themes and subthemes were an accurate representation of the data. NVIVO was used to manage the data.

Analysis indicated that groups of cases (individuals) existed and it was possible to develop a typology. A typology provides a useful way of describing groups of individuals with different clusters of behaviours, values or attitudes. Kluge's[14] four-stage process of construction of empirically grounded types was followed. Stage 1—'development of relevant analysing dimensions'—was achieved through the thematic analysis described above; here, 'dimensions' are themes and subthemes. Stage 2—'grouping of cases and analysis of empirical regularities'—individual cases were compared and contrasted and their position within the themes and subthemes determined. Stage 3—'analysis of meaningful relations and type construction'—relationships between the themes of the groups were analysed and heterogeneity between groups checked, that is, sufficient variation in data existed between the groups. Stage 4—'characterisation of the constructed types'—through an iterative process, including discussion with the wider research team, the final typology of PA maintenance was characterised.

Critical reflection was practised throughout, and peer debriefing was used to explore researcher bias.

### Patient and public involvement

Patients and carers were involved in the conception of this study which forms part of a National Institute for Health Research postdoctoral fellowship. They reviewed the letters of invitation, patient information sheets and consent forms, and suggested amendments. Patients and carers will also be involved in the dissemination of results, supporting the writing of a participant report and using their networks to disseminate results to wider patient communities.

## RESULTS

### Characteristics of respondents

Letters of invitation were sent to 124 individuals who had previously completed one of the four aforementioned programmes. A total of 48 reply slips were returned, yielding a response rate of 39%. Three individuals were found to be ineligible (no diagnosis of GI cancer) and seven could not be contacted. To ensure variation in the sample, purposive sampling of the 38 viable responses was employed considering age, sex, socioeconomic status and level of PA. Twenty-seven individuals were selected for interview.

Participant demographics are described in table 2. Just over half of the sample were male with an average age of 66.3 years, all were white British. The majority (89%) were married and owned their own home (93%), 56% were retired.

### Thematic analysis results

Seven themes were identified, each is set out below with exemplar quotes presented in table 3.

### Disease processes

Participants described the processes of disease, both cancer and comorbidities, and the role this played in relation to their PA participation.

| Table 2 | Demographic characteristics |
|---|---|
| **N=27** | |
| Male | 15 (56%) |
| Age (mean/range) | 66.3 years (41–79) |
| Cancer type | |
| Colorectal | 23 (85%) |
| Oesophageal | 3 (11%) |
| Stomach | 1 (4%) |
| Previous programme (N) | |
| ASCOT | 15 (56%) |
| ENCOURAGE | 4 (15%) |
| Active Everyday | 2 (7%) |
| Move More Northern Ireland | 6 (22%) |
| Ethnicity | |
| White European | 27 (100%) |
| Marital status | |
| Married | 24 (89%) |
| Living with partner | 1 (4%) |
| Single | 1 (4%) |
| Divorced/separated | 1 (4%) |
| Employment | |
| Full time | 5 (18.5%) |
| Part time | 5 (18.5%) |
| Retired | 15 (56%) |
| Volunteering | 2 (7%) |
| Accommodation | |
| Owner occupied | 25 (93%) |
| Rented housing | 2 (7%) |
| Caring responsibilities for adult | 4 (15%) |
| Education | |
| More than secondary education, for example, undergraduate degree | 12 (44%) |

ASCOT, Advancing Survivor Cancer Outcomes Trial; ENCOURAGE, Efficacy of aN exercise training programme during Concurrent neoadjuvant canceR treAtments trial.

### Disease limits activities

For some, late effects of cancer treatment inhibited engagement in certain activities. Limitations as a result of a stoma were described, for example, avoiding activities that involved bending, or swimming due to issues of body esteem and concerns about bag leakage. Some also experienced bowel urgency and described avoiding certain activities where toilet facilities were unavailable, as well as adapting activities due to concerns about leaks or accidents.

Other comorbidities/ill health impacted PA participation, including arthritic conditions and back problems. Chronic obstructive pulmonary disease, asthma and breathlessness were reported to restrict walking outside, particularly in cold weather. More acute periods of illness or injury were also described including cough/colds and muscle/joint strains which interrupted PA participation.

### PA to improve comorbidities/late effects of cancer

In contrast, PA was also described as essential in maintaining good health and well-being, avoiding illness and physical deterioration, and was a key motivating factor for ongoing participation. Participants also recalled the role of PA in alleviating late effects of cancer treatment, for example, exercising daily to improve gut mobility and bowel habit. PA was also reported as important to manage other conditions such as type II diabetes, avoiding weight gain/promoting weight loss and alleviating joint stiffness caused by arthritic conditions. For one participant, engaging in regular PA revolutionised the way he managed chronic back pain (see table 3).

### The role of ageing

Factors related to ageing were frequently referenced in relation to PA participation.

### Ageing perceived as inhibiting PA

In some cases, advancing age was cited as a barrier to engaging in PA. Modification to activities were described such as using the bus rather than walking, as well as a process of 'slowing down', which in some cases was associated with retirement. Some perceived that more structured exercise was not appropriate for older adults.

### PA to combat consequences of ageing

In contrast, participation in PA was also reported to be important to preserve fitness and mobility and slow the physical decline associated with ageing. It was felt that regular exercise helped to ensure a future where the individual can engage in activities they enjoy and honour commitments to family and loved ones for as long as possible. Maintaining independence and avoiding burdening others were also important to some participants. Comparisons were also made to older relatives who they had witnessed decline in health. A desire to avoid such reductions in mobility and independence was expressed.

### Emotion and psychological well-being

The association between PA and psychological well-being and emotion was commonly described, both in a facilitative and deleterious way.

### Psychological well-being

Participants who engaged regularly in PA frequently described enjoying the activities as well as feeling a positive impact on psychological well-being. In some instances, PA also acted as a distraction from concerns about their health and alleviated anxieties. Others spoke of feeling empowered by taking exercise, using it as a vehicle to take control over their health following their cancer diagnosis and treatment.

**Table 3** Themes, subthemes and accompanying quotes

| Theme | Subtheme | Quote |
|---|---|---|
| Disease processes | | |
| | Disease limits activities | *"I used to go swimming but, I don't want to go swimming now because, I don't want to be in changing rooms where I've suddenly got to change the bag and the bag's wet and all—because, I have a—a hernia—I don't want to be wearing swimming costumes with a kind of lump there, you know, a bag lump and a body lump."* Female, ASCOT<br>*"I ought to get more active. I mean, me sister's offered me to. She said: 'oh, yeah', you know: 'I—I go swimming every Tuesday, come with me'. But, I can't think of anything more frightening than if I was taken short in a (swimming pool)…"* Female, Active Everyday<br>*"I used to have a treadmill up to about six months ago and, I was on that most days. Every third day at least. But my balance isn't very good because I had a trapped spinal cord….I half regret getting rid of that…"* Female, Active Everyday |
| | PA to improve comorbidities/late effects of cancer | *"After I'd, you know, got into the—the gym after the exercise very quickly, my condition was transformed beyond all recognition. You know, and that—that was the motivation for keeping it up because I saw what it did for me and, I felt much better. My mobility was much improved and the pain levels had disappeared. I mean, I was on constant pain relief. I was taking Naproxen and Co-codamol. Both. Sort of three or four times a day. Once I'd started into a gym, I was taking none."* Male, Move More |
| The role of ageing | | |
| | Age inhibiting PA | *"I still do me walks but, I don't do nothing like play any—any sport or anything. (laughing) It's a bit much at sixty-nine, I think, you know."* Male, ASCOT |
| | PA to combat ageing | *"If you're fit and healthy obviously it helps, I should think. And, you're not—you're not sitting around and becoming a burden to everyone else. So, this is obviously what we both want to avoid. So, yeah, that's—that's probably the reason behind it (engaging in regular PA) and a fulfilling life, I suppose, and not being a—a burden on anybody for as long as possible (laughing). Up to one-hundred-and-ten at least (laughter)."* Male, ASCOT<br>*"When I look at my father-in-law he's—he's—muscle waste. He's got no strength anywhere. Obviously, arthritis doesn't help. But, I want to make sure I've got the fitness and ability to be able to do more than him when—as I go—go through my older ages. So, that's the one that really sort of kicked in with me and that's why I do cycling and I've got quite strong legs from the running as well."* Male, ASCOT |
| Emotion and psychological well-being | | |
| | Psychological well-being | *"It takes—you know, you're—you're not thinking about that ache, that pain and what's going to happen in the future."* Male, ENCOURAGE<br>*"But, you know, if—I think, it's kind of—it gives you a bit more power, doesn't it?—your power's been taken away from you in that treatment time but—and then you want it back. So, you think, so, I'm taking this back."* Female, ASCOT |
| | Low mood and motivation | **INT: So, going back to you using the exercise bike. Can you pinpoint what it was that stopped you doing that?**<br>*RESP: ….I got to a stage and I'd think: 'why am I bothering?'. I think—I'm getting to a bit of a—down in the dumps stage. I think: 'why am I bothering?'. I'm seventy—seventy-something. I'm seventy-six or whatever. And why should I bother? You know, because I've not got much longer to life. I think, I went through a stage like that.* Female, ASCOT<br>*You go for a walk around the block and once you've been around the block once, you're bored with it. So, then you drive out somewhere and have a walk around and then you run out of places to drive up to and you just can't be bothered.* Male, ASCOT |
| Incorporating PA into everyday life | | |

Continued

**Table 3** Continued

| Theme | Subtheme | Quote |
|---|---|---|
| | Incidental activities | *"I'm pretty active anyway. I mean, I don't—I don't like sitting about too much. I always find something to do."* Male, ASCOT |
| | Planned activities | *"Yeah. Yeah. They (Nordic walking sessions) were sacrosanct. You know, they're in the diary. They're—well, we have a calendar on our desk and my walks are all written in there."* Female, ENCOURAGE |
| Social interaction | | |
| | Avoiding isolation | *INT: And so, it sounds like you get down to that session on a Tuesday as often as you can. What is it about that that keeps you going back week after week? RESP: I think, it's— it's the company. Like, as a—as a pensioner now, I wouldn't—I wouldn't have much contact with other people other than the family. But it's—it's the contact. It's getting out and—and meeting the people which would be the driving force behind it. And as well as that, I enjoy the exercises.* Male, Move More |
| | Friendship | *"Let's put it this way. When we walk our mouths work just as hard as our legs (laughter) so. And—and you—so, you can—you tend to—you know, you walk in pairs usually just because on tracks it'll often—that will be, you know, the— determined by the woods of the track. So, walking two-by-two is the usual and—and you switch around. You know, that's the nice thing. You'll—you'll be in different groups and you just happen to link up with somebody and you'll—you'll have a good old chat as you go along. So, that's the social side bit which is lovely."* Female, ENCOURAGE. |
| Support and self-monitoring | | *"Practice nurses with my GP that—those sort of people lectured me for years about doing a bit more physical activity but they're not offering me anything."* *"We (Active Everyday trainer) talked about whether or not—you know, even when I was younger, I would have played any sport or done anything like that. We— we had a general conversation and that's—that as far as I know is a big part of their role. Is to fit people in with things that they feel you'd be capable of doing because if you weren't capable of doing it you're not going to stick at it."* Female Move More **INT: And so, you talked about buying a Fitbit. How do you find that impacts on what you're doing?** *RESP: Well, it's great because—you know, say I'd got to about mid-afternoon, we're not doing anything and I've only done five thousand steps, I know it's time for a walk. You know, that sort of thing. So, no, I use it as a guide and as a—as a recorder really as well.* Female, Move More |
| Competing demands | | *"And then—then we had—because my husband's an only child, my mother-in-law's got dementia and we were having to look after her and, you know, we had all the worry after—and after she died, clearing the house. So, it's all sort of fell by the wayside, my exercise regime, really which is a shame. But, I was—so, I would say for 2 years, I wasn't sort of doing what I had been doing before when the trial was on. But, I'm trying to gradually get back now."* Female, ASCOT *"So, me cycling, I do every Wednesday, Thursday and Friday and that would be cycling over to pick the grandchildren up. Take them home from school. Look after them for a little while. And then bring them—then come home myself on my bike rather than jump in the car."* Male, ASCOT |

ASCOT, Advancing Survivor Cancer Outcomes Trial; ENCOURAGE, Efficacy of aN exercise training programme during Concurrentneo adjuvant canceR treAtments trial; PA, physical activity.

### Low mood, low motivation and lack of enjoyment

For others, feelings of low mood accompanied apathy and impacted on motivation, leading to a reduction in activity levels. Boredom was also expressed, with some participants struggling to find ways to increase their activity level in a manner that interested them and they enjoyed.

### Incorporating PA into everyday life

Participants described a variety of ways in which they incorporated PA into their lives.

### Incidental activities

Typically, individuals who participated in little or no structured exercise described a desire to avoid long periods of

sitting. Participants recalled engaging in incidental activities such as domestic chores and gardening as a means of staying active.

### Planned activities

For some, engagement was structured, organised and part of their routine, such as attending regular exercise classes or active commuting. Participants expressed commitment to these activities, and for some, engagement was a priority. Others describe periods of increased PA engagement, for example, during the PA programme or prompted by other factors such as a desire to lose weight, which subsequently reduced or stopped. Some participants describe how they re-engaged in activities weeks, months or even years later.

### Social interaction

Engagement in PA provided an important opportunity to socialise.

### Avoiding isolation

For some individuals, living alone and/or who were retired, attending exercise classes provided an opportunity to meet with others and socialise, reducing feelings of isolation.

### Friendship

Others talked about how participation in PA presented opportunities to develop meaningful and long-lasting friendships with those they exercised with. For those participants, engaging in regular PA was an opportunity to spend time with friends and acted as a motivation to continued engagement.

### Support and self-monitoring

For some participants, continued engagement in PA was facilitated by personalised support. This was described by individuals who had previously engaged in regular exercise prior to cancer diagnosis, as well as those who were novices. The one-to-one specialised support provided during involvement in a PA promotion programme helped individuals find activities they enjoyed and were appropriate for them, enabling continued participation. For example, one participant described how a practice nurse 'lectured' her about taking more exercise with little effect. However, following a consultation with the Active Everyday practitioner, she was able to find an activity that suited her needs, and which she had engaged in weekly for more than 2 years. See table 3 for quotes.

Self-monitoring techniques were important for some. The use of technology to monitor activity, particularly steps per day, was described as useful. Participants in the ASCOT trial were asked to wear a pedometer at the beginning of the study. Some reported that they were surprised at how few steps they were doing at baseline, and this provided motivation for change. Some continued to use a device to measure their steps, setting a goal, such as walking 10 000 steps a day and regularly self-monitoring their behaviour.

Further, despite ensuring no advice/recommendations to increase activity were made during the interviews some described their participation in the current study as acting as a prompt to consider re-engaging with PA.

### Competing demands

Responsibilities such as caring for family members, family commitments and work schedules made consistent and continued engagement in PA challenging for some. Others were able to use these commitments as an opportunity to increase their activity levels. For example, playing with grandchildren and active commuting.

### Typology

From these findings describing experiences, barriers and facilitators to long-term PA engagement it was possible to generate a typology consisting of three multidimensional types 'maintained PA', 'intermittent PA' and 'low activity'. This typology, informed by the themes identified, characterises the varied experiences of our sample. This helps to expand our understanding of factors that influence an individual's success (or not) of incorporating regular PA into their everyday lives (see table 4 for characteristics by type and accompanying vignettes).

The first type is '*Maintained PA*' (see figure 1A). Participants described themselves as routinely active, most often engaging in structured moderate and/or vigorous PA at regular times/days. Examples include attending the gym, Nordic walking and group exercise classes. Some had participated in regular PA at other periods of life and the PA intervention supported them to re-engage with these activities. Others established new routines following participation. The defining features of this group include finding enjoyment and pleasure in regular PA which is planned and prioritised. Physical and psychological benefits of regular PA in preserving health, mobility and independence were described and highly valued. Interruptions to activity tended to be brief and resuming activity was not problematic or effortful. PA as a means of socialisation was important for some but not all in this type.

The second type is '*Intermittent PA*' (see figure 1B). Participants in this type described irregular engagement in structured PA, often resulting in cycles of action and inaction. For example, starting an exercise class or purchasing new exercise equipment, participating regularly for a period of time and then subsequently experiencing an interruption to that behaviour. Typically, these participants discussed the known benefits of exercise and described taking part because 'they should', often expressing extrinsically driven motivations such as a desire to lose weight. Reasons for interruptions to PA participation varied and included intrinsic factors, for example, boredom or low mood and extrinsic factors, such as a stressful event, ill health or caring responsibilities. Once these barriers were no longer relevant participants often found it difficult to re-engage due to a lack of motivation/apathy. This group also described deriving less pleasure

**Table 4** Participant characteristics by type

| Type | Age | Sex | Retired % (n) | More than secondary education | Vignette |
|---|---|---|---|---|---|
| Maintained PA n=13 | 41–79 Mean 65.4 | F=43% (n=6) | 57% (8) | 35% (n=5) | Prior to cancer diagnosis, Jean enjoyed regular participation in Nordic walking. She has enjoyed sport/physical activity throughout life but participated less during working life. She describes a sense of calm leading into her operation as she felt she had done all she could to optimise her health through the ENCOURAGE trial. She talks with passion of the importance of continued regular exercise for her health, wanting to be fit enough to enjoy life with her family and grandchildren and champions others to be active, as well as regular exercise giving her a sense of physical and mental well-being.<br>Jean talks about having more time to prioritise exercise in her retirement and states that exercise is at the top of her to-do list each week. Exercise sessions are scheduled into her diary in advance each week and are frequently prioritised over other activities.<br>Jean talks about a love of the outdoors and being active with a particular passion for Nordic walking.<br>Jean walks regularly with a group and enjoys the social element with lasting friendships with other group members. Female, ENCOURAGE trial. |
| Intermittent PA n=5 | 60–76 Mean 65.8 | F=40% (n=3) | 20% (1) | 60% (n=3) | Sue talks about the importance of physical activity to maintain health and describes its importance due to being overweight and experiencing a myocardial infarction.<br>Sue describes periods where she exercised regularly but is currently inactive and says 'I know I should do more'.<br>Following completion of cardiac rehabilitation, Sue purchased a stationary exercise bike, the intention was to use it every day which she reports doing for a few weeks before stopping.<br>Sue describes a lack of enjoyment and little change in weight after using her stationary bike. She describes having low mood and a lack of motivation to continue.<br>Sue also talks about regaining weight she had lost which makes walking and exercising uncomfortable, also reducing her motivation to exercise. Female, ASCOT trial. |
| Low activity n=8 | 61–76 Mean 68.4 | F=50% (n=4) | 75% (6) | 50% (n=4) | Roy describes the presence of a stoma bag 'slowing him down' and engaging in less activity since his retirement.<br>He describes actively seeking opportunities to avoid long periods of sitting and undertakes tasks in and around the home.<br>Roy perceives himself to be sufficiently active compared with others of a similar age.<br>Roy talks about walking to get the paper each day but feels more structured exercise is not appropriate for his age.<br>He also talks about enjoying being able to help family members by completing jobs around the house and does not desire to increase activity levels further.<br>Male, ASCOT trial. |

ASCOT, Advancing Survivor Cancer Outcomes Trial; ENCOURAGE, Efficacy of aN exercise training programme during Concurrent neoadjuvant canceR treAtments trial; PA, physical activity.

and intrinsic reward from engaging in PA than those in the maintained PA type.

The third and final type described is '*Low activity*' (see figure 1C). Participants in this group take part in little or no structured PA but do describe avoiding long periods of sedentary time by engaging in activities of low intensity such as jobs around the home. Reference is also made to incidental activities such as climbing stairs in the home which are felt to maintain mobility and fitness levels. As

with the maintained PA type, the importance of PA to maintain mobility was discussed with those in the lower activity type feeling this was achieved with lower intensity activities such as walking and household chores. Levels of PA participation were felt to be appropriate for their age and perceived capability. Reference was also made to reductions in activity levels compared with earlier in the life. Some are content with this and attribute it as a natural consequence of ageing. Others had to stop or

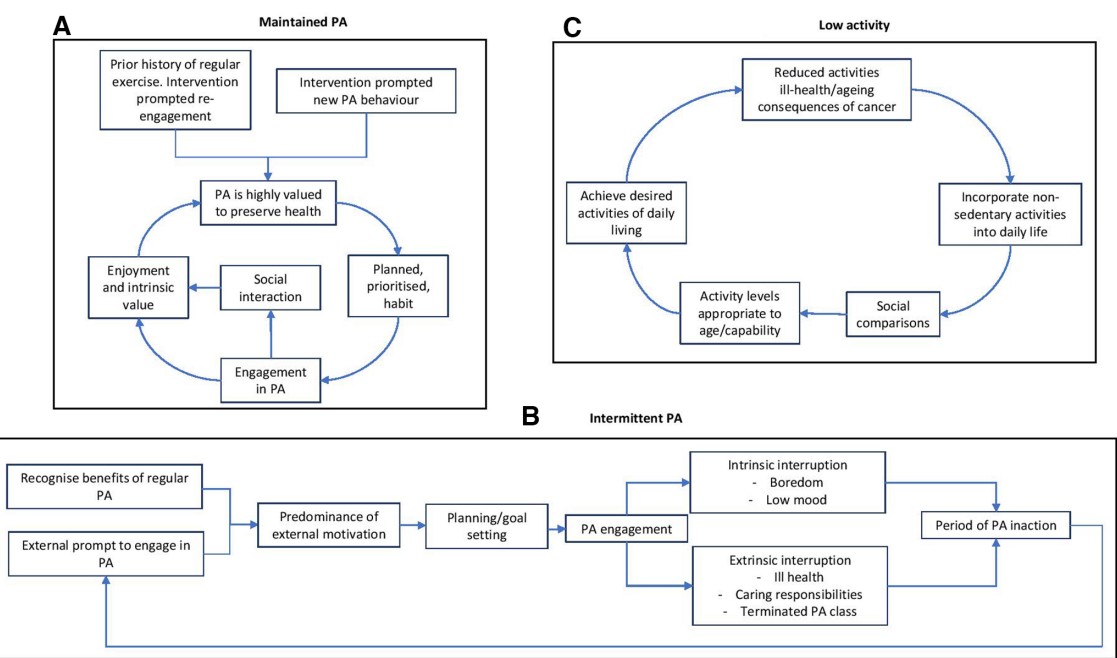

**Figure 1** (A) Maintained PA. (B) Intermittent PA. (C) Low activity. PA, physical activity.

adapt activities less willingly as a result of comorbidities and/or consequences of cancer treatment. Comparisons were also made to others of a similar age with participants evaluating their activity levels to be higher than their peers. Those in the low activity group describe being able to carry out necessary activities of daily living and infrequently express a desire to engage in additional PA. Demographic characteristics of individuals by type (see table 4) show those in the low activity group to be slightly older that those in the other types. There was variation in original intervention participation across all three types.

## DISCUSSION

This study provides the first published insight into the experiences of sustaining long-term PA behaviour change following participation in a PA promotion programme among people LWB GI cancer. Seven key themes were identified, presenting key factors that promote or inhibit regular participation in PA; disease processes, the role of ageing, emotions and psychological well-being, incorporating PA into everyday life, social interaction, competing demands, support and self-monitoring. Additionally, a novel typology was developed to include three types: maintained PA, intermittent PA and low PA.

Characteristics of the 'maintained PA' type, where participation in PA is regular and sustained, are comparable to findings from a systematic review and meta-synthesis of the acceptability of PA interventions in older adults.[15] This review included evidence from studies of older adults (≥65 years old) from non-clinical populations. The authors described the importance of fun and enjoyment as a motivator for engagement, as well as descriptions of functional and psychosocial improvements, related to PA participation which increase perceived value of participation, and

consequently enacted behaviour. Devereux-Fitzgerald *et al* include the two factors of 'enjoyment' and 'value' as central to their analytic model describing PA behaviour as seen in the 'maintained PA' type.[12] The enjoyment of social interaction (as well as of PA participation in and of itself) was also a common theme in the published literature captured in this review and supported in part by the current study. However, social support was not a universal facilitator of continued PA engagement. Other reviews have challenged this assumption that social support is a key facilitator to engagement in PA, reporting it to be important for some but not all.[16 17]

In one of the few qualitative studies investigating long-term PA behaviour in people LWBC, Midtgaard *et al*[18] interviewed individuals who had participated in the Copenhagen PACT study, a 12-month programme to promote PA after cancer. Those interviewed described the importance of prioritising PA and planning ahead to ensure regular engagement. They also described exercise as a prerequisite for 'feeling and staying well'. However, as others have argued,[15] a focus solely on the role of PA for long-term health is unlikely to encourage long-term participation of many older adults. This argument is in keeping with the current study whereby the majority of participants describe knowledge of the value of PA for health but do not necessarily participate regularly.

Barriers identified in this study are consistent with those reported in the existing literature. A recent review of barriers and facilitators to exercise in people LWBC highlighted health problems, work and home responsibilities and lack of motivation as key barriers to engagement.[11] Others describe the deleterious effect of older age, physical decline and lack of time.[16 19] Although most studies in these reviews focused on adoption of PA

rather than maintenance, the few studies that do consider long-term behaviour revealed similar obstacles. Brunet et al[20] interviewed women with a previous diagnosis of breast cancer regarding maintenance of self-directed PA following participation in an intervention. Physical barriers were common, including arthritis and long-term effects of cancer treatment, such as fatigue as well as a lack of motivation, work and caring responsibilities. Similarly, in a study exploring experiences of PA over 5 years since diagnosis, Hefferon et al[21] reported lack of motivation and apathy as well as the deleterious effects of the ageing process and comorbidities and practical barriers such as proximity to exercise facilities and competing priorities as prohibitive to regular long-term PA engagement.

When considering the 'intermittent PA' type with cycles of action and inaction, it is helpful to consider theories of motivation and behaviour change that may help to explain this phenomenon. Kwasnicka et al[22] present a systematic review of behaviour theories with a focus on maintenance of behaviour change and identify five key themes of which self-regulation is one. They propose that individuals differ in their ability to self-regulate their behaviour and those with lower self-regulatory capacity may have weaker intention–behaviour relationships (p. 285). It is suggested that when a behaviour falls short of a relevant goal, for example, an individual does not attend an exercise class as planned, the individual must then exert more cognitive effort to 'correct' this behaviour and bring about satisfaction of goal achievement, or they disengage with the goal. At this stage the new behaviour is effortful, requiring significant self-regulation. Kwasnicka et al[22] describe self-regulation as a limited cognitive resource which can be depleted through things such as stress and illness, and our self-regulatory reserves fluctuate throughout life. Habit formation has been proposed as a strategy to support behaviour maintenance.[23] Habit is a process whereby a cue triggers an impulse to act because a mental association has been learnt between a cue and an action.[24 25] Habits can support maintenance of behaviour because they are controlled through an associative reflexive system, as opposed to an information-processing system,[26] meaning they continue to guide behaviours even when self-regulatory resources are low. Habit formation however can be a long process[27] and maintaining motivation for long enough to develop strong habits can be challenging, therefore long-term behaviour change will only be achieved once the behaviour is habitual, requiring less conscious control. These suggested mechanisms bear many similarities to the narratives presented by those in the intermittent activity type. It is also proposed that motives for sustained behaviour must bring about regular gratification and will be more successful if they are intrinsic, such as the enjoyment described by those in the 'maintained MVPA' type in this study.

A number of the components of the 'low activity' type, characterised by engagement in little or no structured PA, share similarities with evidence from a recent review by McGowan and colleagues who conducted a systematic review and meta-synthesis of the acceptability of PA to older adults.[28] They synthesised the evidence for community-dwelling older adults who had not been involved in PA interventions and were therefore not necessarily active or willing to be active. They describe how older adults in these studies interpreted PA as a 'by-product' of other activities and a perception that more structured and purposeful PA wasn't necessary or appropriate for them. Similar concepts were described in a review of factors influencing PA participation in people with a diagnosis of lung cancer with 'usual activities' preferred over structured exercise.[19]

### Implications for practice and future research

Findings from this study have important implications for practice. In addition to providing novel data of the experiences of maintenance of PA behaviour change in those LWB GI cancer, generation of the unique typology argues for targeted interventions to support long-term behaviour change rather than a 'one size fits all' approach. Participants from all four PA programmes were represented in the three types described here demonstrating successful (or not) behaviour change was not limited to one approach. Those who have a history of exercising and/or find activities that bring them pleasure and hold strong values of the importance of PA will likely need minimal support to sustain engagement. In contrast, individuals who experience cycles of success and failure of engagement need further support with self-regulatory process. More frequent contacts may help them identify appropriate activities, develop action plans and solve problems when they face challenges. However, further research is needed to determine how best to support individuals to self-regulate independently when they experience recurrent cycles of action and inaction in order to prevent reliance on external support.

Raising awareness of the importance and relevance of PA that goes beyond participating in incidental and daily activities is necessary in the low activity type. These individuals may also require support to alter their self-identity, to see participation in PA as an appropriate behaviour for them, which may also be facilitated by alteration of social norms. Further support to find activities appropriate to their physical capabilities, considering comorbidities and long-term consequences of cancer will likely be needed. True for all is the importance of finding activities that are meaningful and enjoyable.

A priority for future research is to develop processes that identify these key characteristics and determine the most appropriate support. Lastly, the striking similarities between themes presented here and those in the wider literature, particularly among clinical and non-clinical populations of older adults, suggest conclusions and therefore recommendations may be appropriate to other cancer types and non-cancer populations.

### Strengths and limitations

Inclusion of a mixed sample consisting of men and women with variation in current activity levels and age is

a strength of this study. Transferability is reduced by the absence of any non-white participants, the vast majority were married or cohabiting and most owned their own home suggesting financial security and a lack of socioeconomic diversity despite attempts to maximise this during sampling. Furthermore, only seven participants were classified as the 'low activity' type and therefore, despite our sampling strategy, are underrepresented in this study.

It is also important to note that we present thematic descriptions of the data and higher level conceptual models in the form of a typology. There are multiple causal mechanisms at play, often beyond the control of the individual which are not described in the typology presented here. Future work should look at mechanisms associated with these types and how these might inform the design of future PA promotion programmes. The authors also note that this typology sits within environmental and structural contexts which were not explored.

## Conclusions

People LWB GI cancer experience numerous challenges to sustaining increases in PA. These include cancer-specific consequences such as presence of a stoma and issues with bowel habit; however, many are similar to those described among non-cancer populations. The novel typology supports the need for stratified approaches to long-term support for PA behaviour change. This approach is in keeping with emerging models of cancer care which call for personalised approaches. Future research should explore whether tailoring of interventions using these types produces better maintenance of PA.

**Author affiliations**
[1]Macmillan Survivorship Research Group, School of Health Sciences, University of Southampton, Southampton, UK
[2]Centre for Clinical and Community Health Psychology, University of Southampton, Southampton, UK
[3]Research Department of Behavioural Science and Health, University College London, London, UK
[4]Faculty of Epidemiology and Population Health, London School of Hygiene and Tropical Medicine, London, UK
[5]NIHR ARC Wessex, School of Health Sciences, University of Southampton, Southampton, UK
[6]College of Nursing, University of South Carolina, Columbia, South Carolina, USA

**Acknowledgements** We would like to acknowledge the work of colleagues who assisted in the identification of participants to this study including Samantha Leggett (ENCOURAGE trial), Liam Humphreys (Active Everyday), Nuala McVeigh, Kelly Irwin, Eimear Hagan and Diarmaid McAuley from Move More Northern Ireland. Finally we would like to thank the participants for their time and invaluable contribution.

**Contributors** CG made substantial contributions to the conception, design, acquisition of data, analysis and interpretation of data for this study. CG drafted the manuscript and approved the final versions. CF made substantial contributions to the conception and design of the study, interpretation of the data, provided critical intellectual content and approved the final version. CRM made substantial contributions to the conception, design of the study, analysis and interpretation of the data, provided critical intellectual content and approved the final version. TC made substantial contributions to the data analysis and interpretation, provided critical intellectual content and approved the final version. BP made substantial contributions to the conception of the design of the study, provided critical intellectual content and approved the final version. MM made substantial contributions to interpretation of the data, provided critical intellectual content and approved the final version. PL made substantial contributions to the acquisition of data, provided critical intellectual content and approved the final version of the manuscript. KB made substantial contributions of interpretation of the data provided critical intellectual content and approved the final version. All authors agree to be accountable for all aspects of the work.

**Funding** This study was funded by a National Institute for Health Research (NIHR), postdoctoral fellowship. The publication presents the independent research funded by the NIHR. The views expressed are those of the authors and not necessarily those of the NHS, the NIHR or the Department of Health and Social Care.

**Competing interests** None declared.

**Patient consent for publication** Not required.

**Ethics approval** Ethical approval was obtained from the Health Research Authority and the University of Southampton Research Ethics committee prior to recruitment and data collection

**Provenance and peer review** Not commissioned; externally peer reviewed.

**Data availability statement** Study data (anonymised interview transcripts) available upon reasonable request following completion of all necessary ethical approvals.

**ORCID iDs**
Chloe Grimmett http://orcid.org/0000-0002-7540-7206
Michelle Myall http://orcid.org/0000-0001-8733-7412

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
