## [Reviewer comments · BMJ Open]

ARTICLE DETAILS

TITLE (PROVISIONAL)	Exploring maintenance of physical activity behaviour change among people living with and beyond gastrointestinal cancer: A cross-sectional qualitative study and typology
AUTHORS	Grimmett, Chloe; Foster, Claire; Bradbury, Katherine; Lally, Philippa; May, Carl; Myall, Michelle; Pinto, Bernardine; Corbett, Teresa

VERSION 1 – REVIEW

REVIEWER	Forberger, Sarah BIPS, Bremen
REVIEW RETURNED	19-Feb-2020

GENERAL COMMENTS	Thank you for the opportunity to review this paper. The topic is of high relevance, especially the focus on long-term maintenance of PA and the inclusion of the habit-forming literature. I have only minor suggestions for revisions. 1) L17: GI should be introduced first.2) L60: Please check the reference (Gardner et al.)3) Within the discussion section you have presented the 3 typologies. The typologies themselves and their description should be part of the results section first before they are addressed in the discussion section.
--

REVIEWER	Liliana Laranjo Australian Institute of Health Innovation, Macquarie University, Australia
REVIEW RETURNED	20-Feb-2020

GENERAL COMMENTS	Very interesting, well-conducted, and relevant study. The authors conducted a qualitative study to explore facilitators and barriers to physical activity after a diagnosis of gastrointestinal cancer. The paper is well-written and the methods are appropriate and described with sufficient detail. The results section is presented in an engaging structure and the discussion adequately interprets the results in the context of the broader literature and existing behaviour change concepts. Study limitations are acknowledged. Minor comments: The abstract should not include citations and the abbreviation GI is not described when it is first used. The conclusion in the abstract does not seem to be entirely based on the study results, particularly the sentence "There is a need to move away from one-size fits all interventions to promoting PA through personalised approaches." I suggest rephrasing.
--

REVIEWER	Christine Friedenreich Alberta Health Services, Canada
REVIEW RETURNED	14-Apr-2020

GENERAL COMMENTS	General Comments: The authors present a cross-sectional qualitative study among 27 colorectal cancer survivors recruited from four different physical activity promotion programmes in Northern England and Ireland that examined the participants' experiences in maintaining their levels of physical activity after completing these programmes. The authors found that the study participants could be grouped into three types who either maintained their levels of physical activity, undertook intermittent physical activity or only achieved low levels of physical activity. They recommend that there is a need to personalize physical activity interventions to ensure maintenance of activity after cancer. The novel contribution of this paper is not well articulated and needs to be more evident throughout (e.g. stating what the literature gap is that they are addressing in the introduction and more clearly delineating what this study adds to the literature in the discussion). Overall, this qualitative study was carefully conducted and described and there were no issues found with either the analysis or manuscript writing which was very clear. The only major comment was the inclusion of verbatim quotes from the study participants' interviews in tables 3 and 4. Since the participants' age, sex, and previous physical activity program were also provided with each quote, it would be possible to identify these individuals. Since patient anonymity is usually required, it seemed surprising to have included these quotes. Furthermore, the quotes did not really add any information that was not already summarized by the themes and sub-themes and described in the text. Hence, a recommendation to remove the quotes is being given here. Specific Comments:  1. Abstract - More detail on the study design should be provided so that it is clear that a cross-sectional, qualitative study was conducted with participants who were previously enrolled in four different PA promotion programs for cancer survivors in northern England and Ireland. Providing this information about the sampling methods used for this study would be very helpful. 2. General recommendation throughout the paper is to add a noun after each use of the word "this" to ensure that the reader knows what "this" is referring to in the previous sentence. 3. Page 4, line 51: "maintenance" is misspelled. 4. Section 3.1, Characteristics of respondents: The authors state that they had 48 respondents of which 38 were viable and then 27 were selected. It is unclear why all 38 respondents were not interviewed since the sample is quite biased being all white British, 56% male and mainly married. Would the diversity of the sample population characteristics have been improved if more participants had been included? 5. Discussion, strengths and limitations: the authors state that a
---

	strength is their inclusion of a large sample that is representative of this disease cohort. This statement needs to be tempered since their sample size was small and the study population was very select. There are concerns with the generalizability of these results that needs to be recognized more fully here.
--	---

REVIEWER	Erika Rees-Punia American Cancer Society
REVIEW RETURNED	10-Jul-2020

GENERAL COMMENTS	This study aimed to describe barriers and facilitators of maintaining PA levels following a structured PA intervention implemented after a GI cancer diagnosis through phone interviews. Authors further classified individuals to create typologies of those remaining active, intermittently active, and low active. Overall, I think this is useful information for developing targeted, personalized interventions, but I feel some clarifications are needed before this paper is suitable for publication.  • There are some details around the participants and interventions in which they previously participated that would be useful to know:  o It appears the four interventions are quite different (for example, one is a distance intervention and three are in-person, they each focus on various stages of the cancer continuum, etc.) and it appears over half of the interview participants came from ASCOT. ♣ It may be helpful to include the original N and N completers for the four interventions in Table 1. ♣ Should there be some discussion around different aspects of the interventions? Is it possible the more 'successful' interventions were more likely to have participants willing to complete an additional study and/or be more likely to remain active? Were the 'maintain PA' type participants more likely to come from one of the interventions?  o Authors state there were common barriers to taking up/participating in PA related to both treatment side effects and cancer-site specific symptoms. This point is clear and understood, but it does make me wonder, since this study only included GI cancer survivors, should the specific cancer site be included in the title ("...in people living with and beyond GI cancer")? Wouldn't survivors of other cancer sites have different sub-themes/general concerns? ♣ I also think it would be useful to have a bit more information on the participants: stage at diagnosis, time since diagnosis, etc. I suspect these factors would also affect PA maintenance/uptake. • I have a few fairly minor concerns about the methods:  o Line 38-9 on page 6 states that authors 'identified eligible' participants, but it is unclear if there are additional criteria beyond prior participation (and completion?) of one of the four interventions. o Was data saturation achieved? o I assume the phone interviews were all one-on-one, but did all 27 participants comment on all the script prompts and/or themes? • I feel that the typology results belong in the results section. I was confused when I did not see them there, but found them- along with new tables/figures- in the discussion. • I was a little confused about the figures and feel they could use a bit more description. For example, do dotted vs. solid lines mean anything? Do the shapes mean anything? They appear to be fairly complex figures and I didn't feel they were explained thoroughly.
---

VERSION 1 – AUTHOR RESPONSE

Reviewer: 1

Reviewer Name: Forberger, Sarah

Institution and Country: BIPS, Bremen

Please state any competing interests or state 'None declared': Non

Thank you for the opportunity to review this paper.

The topic is of high relevance, especially the focus on long-term maintenance of PA and the inclusion of the habit-forming literature. I have only minor suggestions for revisions.

1) L17: GI should be introduced first.

Gastrointestinal written in full in abstract

2) L60: Please check the reference (Gardner et al.)

Since submission of the original manuscript this book chapter has been published and is cited accordingly.

3) Within the discussion section you have presented the 3 typologies. The typologies themselves and their description should be part of the results section first before they are addressed in the discussion section.

We thank the reviewer for this suggestion which was echoed by reviewer 4. We have now included description of the typologies in the results section.

Reviewer: 2

Reviewer Name: Liliana Laranjo

Institution and Country: Australian Institute of Health Innovation, Macquarie University, Australia

Please state any competing interests or state 'None declared': None declared.

Very interesting, well-conducted, and relevant study. The authors conducted a qualitative study to explore facilitators and barriers to physical activity after a diagnosis of gastrointestinal cancer. The paper is well-written and the methods are appropriate and described with sufficient detail. The results section is presented in an engaging structure and the discussion adequately interprets the results in the context of the broader literature and existing behaviour change concepts. Study limitations are acknowledged.

Minor comments:

1. The abstract should not include citations

Citation removed

2. The abbreviation GI is not described when it is first used.

GI now stated in full

3. The conclusion in the abstract does not seem to be entirely based on the study results, particularly the sentence "There is a need to move away from one-size fits all interventions to promoting PA through personalised approaches." I suggest rephrasing.

Thank you for your suggestion, the conclusion has been rephrased: "The typology described here can be used to guide stratified and personalised intervention development and support PA engagement by people living with and beyond cancer".

Reviewer: 3

Reviewer Name: Christine Friedenreich

Institution and Country: Alberta Health Services, Canada Please state any competing interests or state 'None declared': None declared

The authors present a cross-sectional qualitative study among 27 colorectal cancer survivors recruited from four different physical activity promotion programmes in Northern England and Ireland that examined the participants' experiences in maintaining their levels of physical activity after completing these programmes. The authors found that the study participants could be grouped into three types who either maintained their levels of physical activity, undertook intermittent physical activity or only achieved low levels of physical activity. They recommend that there is a need to personalize physical activity interventions to ensure maintenance of activity after cancer.

1. The novel contribution of this paper is not well articulated and needs to be more evident throughout (e.g. stating what the literature gap is that they are addressing in the introduction and more clearly delineating what this study adds to the literature in the discussion).

Thank you. We agree with the reviewers comment and feel the edits made to more explicitly state the novel contribution of this paper, in both the introduction and discussion have strengthened the manuscript.

Overall, this qualitative study was carefully conducted and described and there were no issues found with either the analysis or manuscript writing which was very clear.

2. The only major comment was the inclusion of verbatim quotes from the study participants' interviews in tables 3 and 4. Since the participants' age, sex, and previous physical activity program were also provided with each quote, it would be possible to identify these individuals. Since patient anonymity is usually required, it seemed surprising to have included these quotes. Furthermore, the quotes did not really add any information that was not already summarized by the themes and sub-themes and described in the text. Hence, a recommendation to remove the quotes is being given here.

Thank you for this observation regarding risk to anonymity. We have now edited to include only sex and study. We feel that the quotes are important to bring the themes and sub-themes to life and, given the novelty of the study, feel they are important to include. It is also a specification of the Consolidated criteria for Reporting qualitative research (COREQ) checklist (point 29) which we include with our submission.

Specific Comments:

1. Abstract - More detail on the study design should be provided so that it is clear that a cross-sectional, qualitative study was conducted with participants who were previously enrolled in four different PA promotion programs for cancer survivors in northern England and Ireland. Providing this information about the sampling methods used for this study would be very helpful.

Thank you for this suggestion, we have included further details on the sampling methods in the abstract.

2. General recommendation throughout the paper is to add a noun after each use of the word "this" to ensure that the reader knows what "this" is referring to in the previous sentence.

Thank you for this suggestion, we have edited throughout the paper as suggested.

3. Page 4, line 51: "maintenance" is misspelled.

Thank you for highlighting this typographic error which has been corrected.

4. Section 3.1, Characteristics of respondents: The authors state that they had 48 respondents of which 38 were viable and then 27 were selected. It is unclear why all 38 respondents were not interviewed since the sample is quite biased being all white British, 56% male and mainly married. Would the diversity of the sample population characteristics have been improved if more participants had been included?

We agree with the reviewer that there is a lack of ethnic diversity in the included sample. All 38 respondents were white British, and the vast majority were married. We were unable to interview all respondents due to capacity issues and had sufficient information power not to require these. As described in section 3.1, the 27 included participants were purposefully sampled from the 38 to maximise diversity of age, socioeconomic status and activity level.

5. Discussion, strengths and limitations: the authors state that a strength is their inclusion of a large sample that is representative of this disease cohort. This statement needs to be tempered since their sample size was small and the study population was very select. There are concerns with the generalizability of these results that needs to be recognized more fully here.

Thank you for this suggestion. The strengths and limitations section has been edited to remove reference to the sample being representative of the disease cohort. Reflections on the limitation of generalisability are also stated.

Reviewer: 4

Reviewer Name: Erika Rees-Punia

Institution and Country: American Cancer Society Please state any competing interests or state 'None declared': None declared

This study aimed to describe barriers and facilitators of maintaining PA levels following a structured PA intervention implemented after a GI cancer diagnosis through phone interviews. Authors further classified individuals to create typologies of those remaining active, intermittently active, and low active. Overall, I think this is useful information for developing targeted, personalized interventions, but I feel some clarifications are needed before this paper is suitable for publication.

There are some details around the participants and interventions in which they previously participated that would be useful to know:

1. *It appears the four interventions are quite different (for example, one is a distance intervention and three are in-person, they each focus on various stages of the cancer continuum, etc.) and it appears over half of the interview participants came from ASCOT. It may be helpful to include the original N and N completers for the four interventions in Table 1.*

We agree with the reviewer that this information would be helpful however it is not available for either Active Everyday or Move More Northern Ireland which are ongoing community programmes.

2. *Should there be some discussion around different aspects of the interventions? Is it possible the more 'successful' interventions were more likely to have participants willing to complete an additional study and/or be more likely to remain active? Were the 'maintain PA' type participants more likely to come from one of the interventions?*

We thank the reviewer for this suggestion and have included a reflection on 'type' allocation and original programme completed in the results section. There was variation across the types in this regard. We also agree that there was a risk that those who agreed to be interviewed for this second study may have higher levels of PA which is why we purposefully sampled to include representation from those with lower activity levels. Despite these efforts only 7 were classified as the 'low activity' type and we have included a reflection on this in the results and discussion.

3. *Authors state there were common barriers to taking up/participating in PA related to both treatment side effects and cancer-site specific symptoms. This point is clear and understood, but it does make me wonder, since this study only included GI cancer survivors, should the specific cancer site be*

included in the title (“...in people living with and beyond GI cancer”)? Wouldn’t survivors of other cancer sites have different sub-themes/general concerns?

We agree with the reviewer that it would add clarity to include reference to the specific cancer population under study in the title and have added this accordingly.

4. I also think it would be useful to have a bit more information on the participants: stage at diagnosis, time since diagnosis, etc. I suspect these factors would also affect PA maintenance/uptake.

We agree with the reviewer that this information would be of interest to the reader but unfortunately, we do not have these data as we did not have ethical approval to access medical details.

I have a few fairly minor concerns about the methods:

5. Line 38-9 on page 6 states that authors ‘identified eligible’ participants, but it is unclear if there are additional criteria beyond prior participation (and completion?) of one of the four interventions.

The eligibility criteria are described in the line above. Individuals with other cancer types took part in 3 of the 4 original programmes. We have restructured this section to improve clarity.

6. Was data saturation achieved?

There is continuing debate in the literature regarding data saturation in qualitative research. Although it is frequently reported in manuscripts that data saturation has been reached, this is typically stated without specifying how this was assessed (Malterud et al., 2015). Data saturation is a requirement of grounded theory studies (which this study was not). Furthermore, as argued by Malterud et al (2015), saturation requires a study that, if not longitudinal in design, has at least employed the constant comparative method proposed by Barney Glaser which our study did not. Our paper reports on a single-phase study that has focused on the identification, characterization, and explanation of differences in a small corpus of data and thus ‘information power’ as presented by Malterud et al is a more appropriate guide to sample size calculation. The study design and analysis are cross-sectional and interpretive and have revealed clearly defined variations in orientation to participants’ attributions about the normative expectations that are held up to them, and their accounts of the actions that they have taken in response to these as well as the constraints that are placed on these. Therefore, it has been possible for us to develop a typology with three different types from a relatively small body of data and we do not claim achievement of ‘data saturation’ but are confident in our ‘information power’.

Malterud et al., Sample size in qualitative interview studies: guided by information power. 2015. Qualitative Health Research 1-8 DOI: 10.1177/1049732315617444

7. I assume the phone interviews were all one-on-one, but did all 27 participants comment on all the script prompts and/or themes?

Yes, all phone interviews were one-to-one (this has been clarified in the manuscript). All main items listed in the interview guide were asked of all participants though not necessarily in the same order.

8. I feel that the typology results belong in the results section. I was confused when I did not see them there, but found them- along with new tables/figures- in the discussion.

We thank the reviewer for this suggestion which was echoed by reviewer 1. We have now included description of the typologies in the results section.

9. I was a little confused about the figures and feel they could use a bit more description. For example, do dotted vs. solid lines mean anything? Do the shapes mean anything? They appear to be fairly complex figures and I didn’t feel they were explained thoroughly.

We thank the reviewer for this reflection, the response to which adds clarity to the paper. We have replaced dotted lines with solid lines and improved consistency of arrow and box shapes to ease interpretation of the figures. A description of each type to accompany the figure is now in the results section.

VERSION 2 – REVIEW

REVIEWER	Christine Friedenreich Alberta Health Services
REVIEW RETURNED	07-Aug-2020

GENERAL COMMENTS	The authors have fully addressed all of the concerns and comments that I raised in the initial review and I have no further suggestions.
--

REVIEWER	Erika Rees-Punia American Cancer Society, USA
REVIEW RETURNED	29-Jul-2020

GENERAL COMMENTS	The authors did a nice job responding to reviewer comments and the manuscript has improved in clarity. One last thought would be to double check the numbers—Page 29 says ‘only 7 participants were classified as low activity’, but Table 4 says N = 8 for low activity. Further, in Table 4, one person appears to be unaccounted for (N = 13 + N = 5 + N = 8 includes only 26/27 interviewed). I would also consider adding lack of information on stage at diagnosis, time since diagnosis, etc. in the limitations.
--